# Stable isotope (C, N, O, and H) study of a comprehensive set of feathers from two *Setophaga citrina*

**Samiksha Deme**[ID]*, **Laurence Y. Yeung**[ID], **Tao Sun, Cin-Ty A. Lee**

Department of Earth, Environmental and Planetary Sciences, Rice University, Houston, Texas, United States of America

* srd8@rice.edu

**Data Availability Statement:** All relevant data are within the Supporting information files.

**Funding:** The authors received no specific funding for this work.

## Abstract

Oxygen, hydrogen, carbon and nitrogen stable isotopes were measured on a comprehensive sampling of feathers from two spring Hooded Warblers (*Setophaga citrina*) in Texas to evaluate isotopic variability between feathers and during molt. Isotopic homogeneity within each bird was found across all four isotopic systems, supporting the hypothesis that molt in these neotropical migrants is fully completed on the breeding grounds. This homogeneity suggests that the isotopic composition of a single feather is may be representative of the whole songbird. However, each bird was found to have one or two outlier feathers, which could signify regrowth of lost feathers after prebasic molt.

## Introduction

The isotopic analysis of bird feathers has been used to infer patterns of diet, foraging, migration, and other ecological descriptors that characterize the life histories of individual organisms [1–4]. Carbon and nitrogen isotopes, in particular, are used to construct an understanding of the dietary niches [5]. For example, the $^{13}C/^{12}C$ ratios in feathers reflect the composition of vegetation in the area of feeding due to differences in plant photosynthetic pathways [6, 7]. In addition, $^{15}N/^{14}N$ ratios in feathers reveal information about a bird's trophic level [8]. Oxygen and hydrogen isotopes in precipitation are strongly linked to local hydroclimate, so $^{18}O/^{16}O$ and D/H ratios in feathers have been used to reconstruct migratory pathways [9–11].

Most of the above studies have been performed on feathers sampled from live birds. To prevent unnecessary harm to the bird, isotopic analyses are usually done on single feathers. While there have been studies on the feathers of deceased birds, such as the use of museum specimens to examine the long-term evolution of stable-isotope composition of the Eastern Whip-poor-will, (*Antrostomus vociferus*), a complete set of feathers is rarely analyzed to study diet and migration [12].

Many birds are thought to undergo molt while on the breeding grounds, so it is commonly assumed that environmental and ecological factors are constant over the course of the molt [13]. If correct, single-feather analysis would be justified. However, natural variability between

**Competing interests:** The authors have declared that no competing interests exist.

feathers, even on birds that undergo complete molt before migration, has not been fully evaluated. Furthermore, it has been suggested that some birds molt during post-breeding dispersal or even during the early stages of migration [13]. In such instances, a detailed understanding of the molt sequence is a necessary prerequisite for reconstructing a bird's ecological behavior and migratory pattern from individual feathers [14]. A better understanding of molt strategy is also important for the study of other cyclical processes related to molt, such as migration and breeding [13, 14].

What is known about avian molt strategy is usually gleaned through observations or measurements of individual birds, in either natural or laboratory settings, or inferences from a species' evolutionary history [13, 14]. This approach, however, requires numerous observations because birds photographed or captured in the field provide only instantaneous snapshots of molt, which must be combined in aggregate to reconstruct a species' entire molt history. A complete set of data is rarely available.

Graves et al. (2018) [15] conducted an analysis of D/H ratios for 24-feather sets (9 primaries, 6 secondaries, 6 rectrices, and 3 patches of ventral contour feathers) from specimens of the black-throated blue warbler (*Setophaga caerulescens*). That study focused on the deuterium isotope composition in territorial male specimens to analyze variability generated during the complete annual prebasic molt. Results showed significant within-individual variations (i.e., $\delta D$ values ranging from 12 to 60‰) across all pterolographic variables in the feather sets, as well as large interannual variability in both absolute $\delta D$ values and their trends within feather classes.

Motivated in part by the Graves et al. (2018) [15] study, we use the stable isotope approach to reconstruct molt strategies and timing in two AHY (after hatch year) male Hooded Warblers (*Setophaga citrina*). In particular, we examined the isotopic variability between feathers to assess the robustness of analyzing individual feathers to infer a specimen's ecological history. We conducted a comprehensive stable isotope analysis (i.e., their $\delta^{13}C$, $\delta^{15}N$, $\delta^{18}O$, and $\delta D$ values) of feathers from two window-kill *S. citrina* specimens from April 2017 and March 2018. Hooded Warblers breed in the southeastern United States and further north along the eastern margin of the continent to southern New York [16]. They winter along the Atlantic coast of Central America and the Caribbean islands [16] (Fig 1). Like most songbirds, they are thought to complete molt on their breeding grounds in summer, allowing us to test the null hypothesis that Hooded Warbler feathers should show limited isotopic variability. This study provides a baseline for interpreting the isotopic compositions of their feathers.

## Methods

Body feathers and a complete set of flight feathers were sampled from two AHY male Hooded Warblers collected on April 2, 2017 and March 28, 2018. These specimens were window-kills on the campus of Rice University in Houston, Texas, USA (29.7˚ N, 95.4˚ W).

The following feather samples from each bird were collected and labeled (Fig 2): primaries P1-P9, secondaries S1-S9, right (RR) and left (LR) rectrices 1–6 (as seen from dorsal view), the primary, median, and greater coverts (PC, MC, and GC), and aggregate body feathers from the back, rump, throat, nape, breast, and belly. The feathers were cleaned in a solution of 2:1 diethyl ether:methanol and suspended in an ultrasonic bath for 2 three-minute cycles to ensure the removal of contaminants and organic detritus on the feathers' surface, following the technique of Bontempo et al. (2014) [17]. Samples were allowed to air dry for 48 hours in glass tubes before sub-sampling for isotopic analysis.

Feather sub-samples were analyzed for bulk isotope composition (i.e., $^{13}C/^{12}C$, $^{15}N/^{14}N$ and $^{18}O/^{16}O$ and D/H ratios reported as $\delta^{13}C$, $\delta^{15}N$, $\delta^{18}O$ and $\delta D$ values, respectively) on a

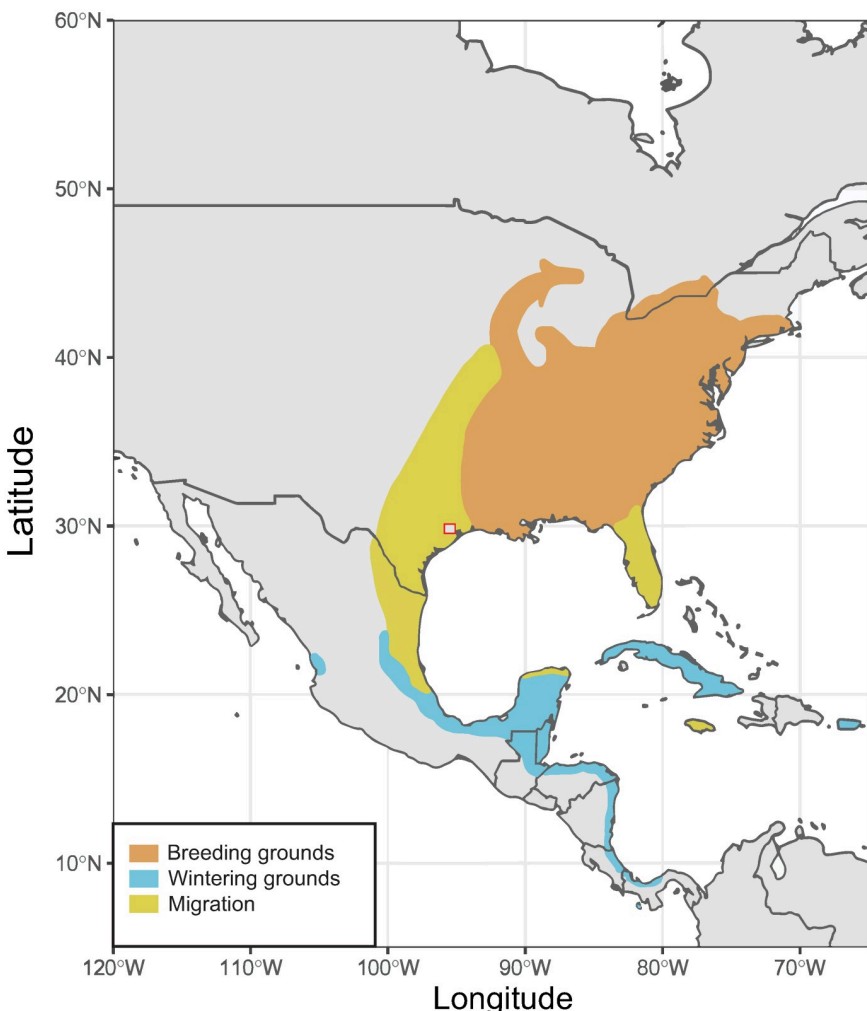

**Fig 1. Map of the migration grounds for *S. Citrina*.** Breeding takes place through June-July while nonbreeding takes place December-February. Migration takes place in Aug-Oct and Mar-May [16]. Square indicates the location of sample collection in Houston, Texas.

ThermoScientific Flash HT Plus-Delta V Plus Elemental Analyzer-isotope ratio mass spectrometry system (EA-IRMS) in the Department of Earth, Environmental and Planetary Sciences at Rice University. For each analysis, new sections of the feathers were cut from the upper section of the feather, placed in a capsule, and weighed. In sub-sampling, we did not separate the rachis from the vane, so our measurements represent the bulk feather segment. Because we sampled the upper portion of the feather, no samples included the calamus.

We believe our bulk analysis of the upper segment of feathers is appropriate for the purposes of this study. Gordo (2020) [18] showed no isotopic gradient between the bottom and tip for passerine feathers, which allows for sampling to occur from anywhere along the feather. Bontempo et al. (2014) [17] found homogeneous isotopic values for the rachis and vane when the calamus was excluded from analysis. However, isotopic fractionation between the vane and the rachis has been observed. Gordo (2020) [18] showed that the rachis and vane were slightly different in δD values, but their isotopic compositions were highly correlated. In our study, we bulk sampled the feathers, resulting in relatively constant proportions of rachis to vane. This systematic sampling approach, combined with the fact that the rachis dominates the

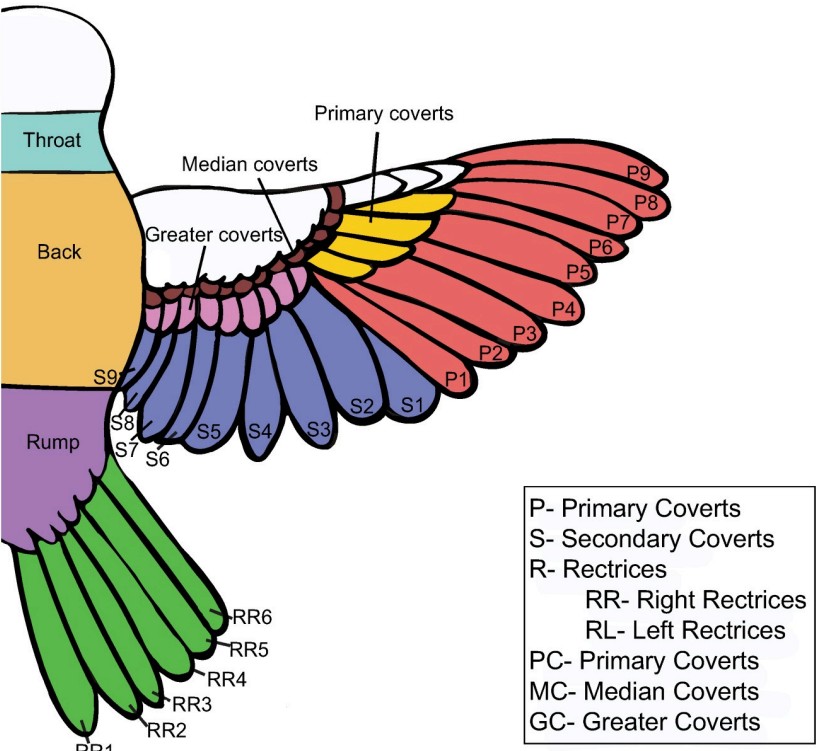

**Fig 2. Diagram of feathers analyzed.** Primary [P] and secondary [S] flight feathers are shown in red and blue respectively, along the right [RR] and left [RL] rectrices in green. The primary coverts [PC] are in yellow, the median coverts [MC] are in brown, and the greater coverts [GC] are in pink. The dorsal body feathers are also shown, with rump feathers shown in purple, back in orange, and throat in teal.

total feather mass, ensures that any isotopic variability between feathers in our study is not due to artifacts of biased sampling of vane or rachis material [13, 14, 18, 19].

For the analysis of $\delta^{15}N$ values, 0.30 mg samples of each feather were put into a tin capsule and analyzed using EA-IRMS. The ammonium sulfate standards IAEA-N1 ($\delta^{15}N = 0.4‰$) and IAEA-N2 ($\delta^{15}N = 20.3‰$) were used for calibration. For the analysis of $\delta^{13}C$ values, 0.20 mg samples were measured, and the standards IAEA-CH-7 (polyethylene; $\delta^{13}C = -32.151‰$) and IAEA-603 (calcite; $\delta^{13}C = 2.46‰$) were used for data calibration. For $\delta^{18}O$ values, 0.10 mg of samples were measured along with the Caribou Hoof standard (CBS; $\delta^{18}O = 3.8‰$) and Kudu Horn Standard (KHS; $\delta^{18}O = 20.3‰$) for calibration.

For $\delta D$ values, 0.20 mg samples were measured along with Caribou Hoof standard ($\delta D = -137‰$) and Kudu Hood Standard ($\delta D = -35‰$). Weighed sample feathers and the standards CBS and KHS (both keratin standards) were put in loosely wrapped tin capsules, which allowed for re-equilibration of exchangeable hydrogen with ambient air moisture [20, 21]. The equilibration took place from December to January (the next year) at a room temperature at ~23°C and a relative humidity of 40–50%. Under these conditions, complete isotopic equilibration should occur by the end of the 15 to 20-day period [22]. The hydrogen isotopic composition of lab air moisture appeared relatively stable during the winter based on the reproducibility of raw $\delta D$ values for CBS between batches measured over an eight-week period (±4‰; 1σ). Because lab air-equilibrated keratin standards are also the isotopic anchors for calibration, the measured isotopic variations in the feathers are due to the non-exchangeable hydrogen isotope fraction.

$\delta^{15}N$, $\delta^{13}C$, $\delta^{18}O$, $\delta D$ values are reported with respect to international standards air $N_2$ ($\delta^{15}N$), VPDB ($\delta^{13}C$) and VSMOW ($\delta^{18}O$ and $\delta D$), and the analytical precisions are ±0.3‰, ±0.2‰, ±0.4‰, and ±5.0‰ respectively (1σ). Standards for all analyses were measured six times during each sample run. Some sample analyses were performed in duplicate or triplicate—depending on sample weight availability, particularly for any anomalous values (see below)—to evaluate external reproducibility.

## Results

The isotopic compositions for the 2017 and 2018 birds are compared in Fig 3. The $\delta^{15}N$, $\delta^{13}C$, $\delta^{18}O$, and $\delta D$ values (Fig 3) show limited variability within each bird except for anomalous values associated with one or two feathers of each bird. The 2017 sample contained just one reproducibly anomalous isotopic composition—the $\delta^{15}N$ value of the back feathers—while the 2018 sample contained three reproducible isotopic anomalies: the $\delta^{13}C$ and $\delta^{15}N$ values of the RL2 feather ($p < 0.0002$ for both) and the $\delta^{18}O$ value of the S3 feather ($p = 0.004$). Replicate analyses of these anomalous feathers ($n = 2$ or $3$ depending on available sample weight) suggest that the anomalies are not analytical artifacts.

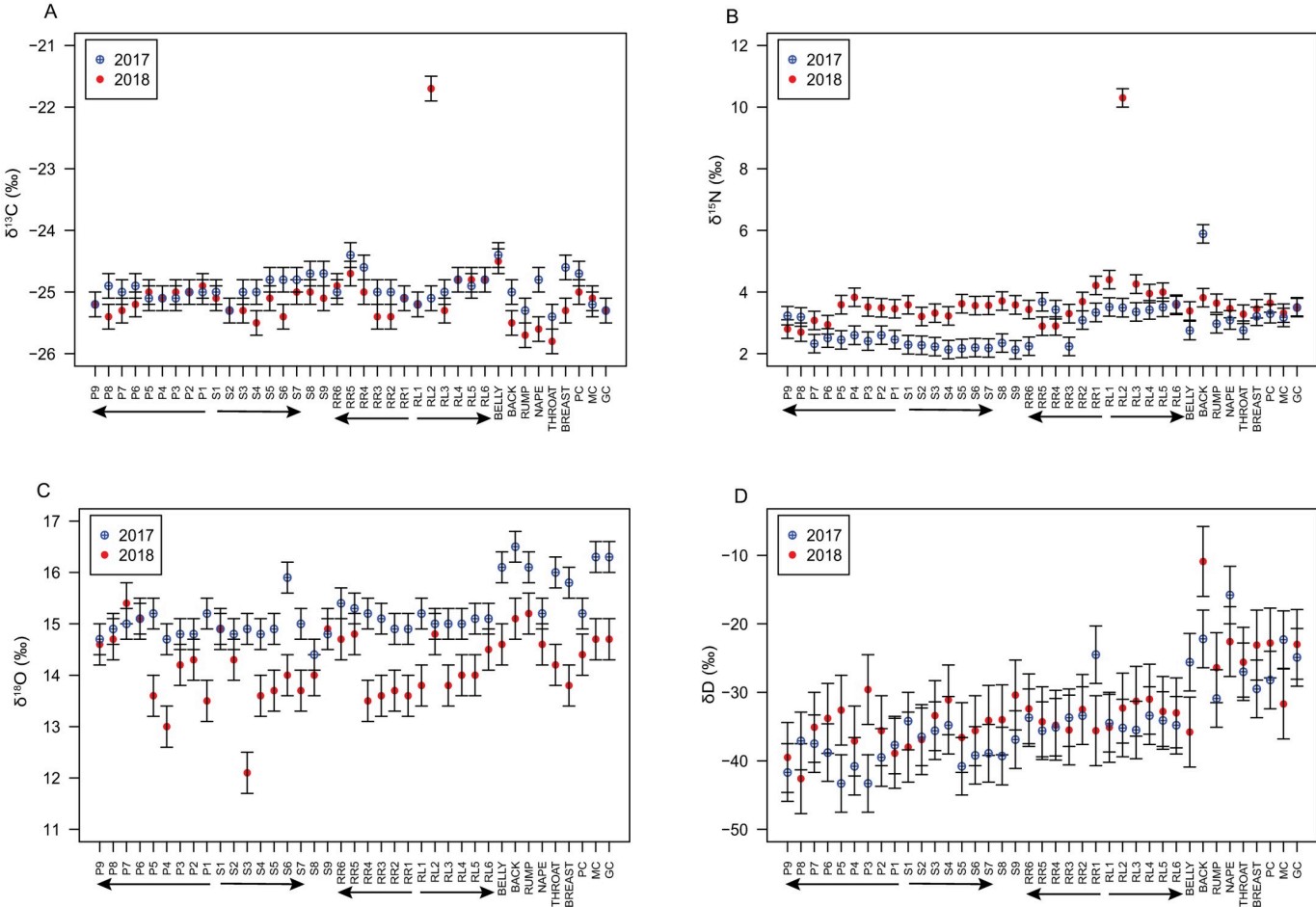

**Fig 3. Comparison of the feather $\delta^{13}C$, $\delta^{15}N$, $\delta^{18}O$, and $\delta D$ values between the 2017 and 2018 samples.** Statistical differences between the samples are shown in Table 1. The 2017 sample values are shown in blue, crossed circles while the 2018 samples are shown in red, full circles. Arrows below the x-axis show the proposed molt strategy from earliest feather molted to last feather molted for each section [13].

No statistically significant difference was observed between the mean $\delta^{13}$C values of the feathers between the birds ($p$ = 0.23), but a statistically significant difference was observed between the birds in their mean $\delta^{15}$N values ($p$ = 0.0004), mean $\delta^{18}$O values ($p$ = $3 \times 10^{-11}$) and mean $\delta$D values ($p$ = 0.045). The mean $\delta^{15}$N, $\delta^{18}$O, and $\delta$D values of the 2018 bird were 0.8 ± 0.1‰ higher, 1.0 ± 0.1‰ lower, and 1.8 ± 0.6‰ higher (1 s.e.m.), respectively, than those of the 2017 bird.

We also evaluated whether isotopic signatures varied as a function of the hypothetical molt schedule proposed in Howell (2010) [13]. These hypothesized molt schedules, from the first to the last feather molted in the sequence, were P1 → P9, S1 → S7 and S1 → S9, RL1 → RL6, and RR1 → RR6. Both the secondary molt schedules S1 → S7 and S1→ S9 were evaluated because the molt schedule of S8 and S9 can occur after that of the other secondaries. A significant correlation, replicated between birds and across isotope systems, would be taken as support for these molt schedules. However, a lack of correlation does not necessarily rule them out; a less protracted molt or a complete molt in a single location would yield uncorrelated isotopic compositions for these feather sequences. Calculated correlation coefficients in each isotopic system are shown in Table 1. While some of the correlations are significant at the 95% confidence level on the 2017 bird (e.g., RR1 → RR6 for $\delta^{18}$O), no such correlations are seen for the 2018 bird. Similarly, high correlation coefficients in one sample for the for $\delta^{13}$C values of flight feathers and left rectrices were not seen in the other sample.

## Discussion

The molt pattern for *S. citrina* is hypothesized to be the primaries from P1 to P9 distally, the secondaries proximally from S1 to S7, the tertiaries distally from S7-S9, and the rectrices distally from the central RL/RR1 to RL/RR6 [13]. The location of the molt is thought to take place primarily during the months of June and August on the breeding grounds. The data show no reproducible pattern in the isotopic compositions of the feathers that resembles the proposed pattern of feather replacement. These inconsistent correlations with the molt sequences hypothesized above support the view that *S. citrina* undergoes complete molt at one location. Furthermore, no significant systematic isotopic fractionations within flight feathers and rectrices was found. We note, however, that body feathers on both birds have slightly higher $\delta^{18}$O and $\delta$D values compared to the flight feathers and rectrices (Fig 3). It is not clear if these systematic differences are related to slightly different isotopic fractionation during the growth of body feathers or slightly different molt locations for the body feathers (i.e., in the more southerly wintering grounds). Both possibilities should be explored in more detail with *S. citrina* samples from a known molt location. In any case, isotopic fingerprinting using single feathers of *S. citrina* should be robust if flight feathers or rectrices are used.

**Table 1. Correlation coefficients ($R^2$) for feather isotopic compositions involved in several proposed molt patterns.**

|  | P1 → P9 | | S1 → S9 | | S1 → S7 | | RL1 → RL6 | | RR1 → RR6 | |
|---|---|---|---|---|---|---|---|---|---|---|
|  | 2017 | 2018 | 2017 | 2018 | 2017 | 2018 | 2017 | 2018 | 2017 | 2018 |
| $\delta^{13}$C | 0.0059 | 0.77 | 0.73 | 0.1805 | 0.62 | 0.0065 | 0.819 | 0.072 (0.65)* | 0.24 | 0.51 |
| $\delta^{15}$N | 0.39 | 0.61 | 0.080 | 0.33 | 0.51 | 0.16 | 0.11 | 0.24 (0.87)* | 0.046 | 0.51 |
| $\delta^{18}$O | 0.067 | 0.33 | 0.0075 | 0.020 | 0.23 | 0.054 | 0.0070 | 0.0099 | 0.91 | 0.63 |
| $\delta$D | 0.0074 | 0.13 | 0.36 | 0.34 | 0.58 | 0.13 | 0.081 | 0.12 | 0.49 | 0.18 |

Molt strategies were hypothesized by Howell (2010) [13] based off observations of *S. citrina* in the wild.

*Values in parentheses are computed without the outlier values.

The mean $\delta^{13}$C values were not statistically different between birds. The similarity in $\delta^{13}$C values suggests that the birds consumed insects that fed on plants utilizing similar photosynthetic pathways (e.g., C4, C3, CAM) [4, 7]. To evaluate this hypothesis more quantitatively, we use diet enrichment factors. The diet of the hooded warbler is primarily insects, such as caterpillars, moths, flies, and other arthropods on breeding grounds [16]. While $\delta^{13}$C enrichment factors in *S. citrina* are not known, Mizutani et al. (1992) [23] showed that the average $\delta^{13}$C enrichments across 11 adult species of birds is 2.5–3.8‰. Chamberlain et al. (1997) [22] found a similar $\delta^{13}$C enrichment of 3–4‰ in black throated blue warblers compared to local C3 plants. Isotopic compositions in insects can vary depending on species and their diet, but because most of the insects in the hooded warbler diet are herbivorous, we estimate the mean $^{13}$C enrichment of the insects relative to local plants using the general fractionation for whole consumers, i.e., $\Delta\delta^{13}$C = 0.5 ± 0.13‰ [24]. Thus, the mean feather $\delta^{13}$C value of -25.0‰ suggests that the environment supporting *S. citrina* during molt had plants with mean $\delta^{13}$C values near -29‰, which is similar to the mean isotopic composition of C3 plants in temperate ecosystems ($\delta^{13}$C ~ -28.5‰) [25].

$\delta^{15}$N values in biological tissues are often interpreted to reflect the trophic levels of food sources; thus, the slight but significant differences between the two birds is noteworthy. Plant matter of the southern and eastern United States and Neotropics has an average $\delta^{15}$N value slightly less than zero, and Mizutani et al. (1992) [23] showed an average $^{15}$N enrichment factor of 3.7–5.6‰ for 11 adult bird species [26]. The average $\delta^{15}$N value of 2.8‰ and 3.6‰ for the 2017 and 2018 birds, respectively, is therefore consistent with the diets of both birds consisting primarily of herbivorous insects [27]. However, the higher $\delta^{15}$N values of the 2018 bird imply that it consumed more insects that were, on average, slightly higher in trophic level (i.e. non-herbivorous) than the insects consumed by the 2017 bird [28]. Nevertheless, the $\delta^{15}$N values for both samples are consistent with the known diet of *S. citrina* and could potentially be used to identify differences in diet between different birds. However, they cannot distinguish between temporal differences in molt.

Chamberlain et al. (1999) [22] and Hobson et al. (2004) [1] found that the deuterium isotopic composition of migratory bird species was 10–30‰ lower in feathers compared to local meteoric waters. The average feather $\delta$D values of -34 ± 6‰ and -32 ± 6‰ (1$\sigma$) for the 2017 and 2018 birds, respectively, therefore suggests that local meteoric waters during molt had $\delta$D values of 0 to -20‰. These values fall within the range of meteoric waters for the southeastern portion of the breeding grounds for *S. citrina* in the southeastern United States [29]. The full range of $\delta$D values in the feathers is consistent with waters extending into the northern region of the breeding range as well.

While the range of $\delta$D values for the full feather set showed some variability, excluding the body feathers shows relative homogeneity for the flight feathers. This contrasts with the findings of Graves et al. (2018) [15], which found $\delta$D variation within flight feathers for *S. caerulescens*. There are a few possible explanations for this discrepancy. One explanation could be methodology: Graves et al. (2018) [15] analyzed only the vane of the feathers, whereas both the vane and rachis were analyzed in this study. In addition, differences in the habitats, diets, and/or isotopic fractionation between the two species could also lead to detectable differences in isotopic trends. For example, while both species are insectivorous, *S. caerulescens* feeds over a wider diversity in structural height, from understory to lower canopy, whereas *S. citrina* primarily feeds in the understory. *S. caerulescens* also inhabit a wider altitude range (roughly 790–1,600 m in the summer months and close to sea level during wintering months) while *S. citrina* lives up to 1,100 m year-round; because $\delta$D values of precipitation decrease nonlinearly with altitude, the wider altitude range of *S. caerulescens* may also contribute to its larger isotopic variability [16, 30, 31].

The magnitude of metabolic fractionation of $^{18}$O remains unclear, due to the larger number of sources and sinks for oxygen compared to hydrogen relevant for feather growth [1]. However, the δ$^{18}$O values of the feathers may provide information about the molt location through a correlation with δ$^{18}$O values in meteoric water [1, 10], similar to how δD values of the feathers can provide information about the molt location [1]. However, laboratory studies of Japanese quails and house sparrows show that the slope of the relationship between drinking water and δD can vary by species [32, 33]. In this study, the relationship between δ$^{18}$O and δD values for both samples neither followed the global meteoric water line (i.e., δD = 8*δ$^{18}$O + 10) nor local meteoric water lines in the breeding or wintering regions, perhaps due in part to the narrow range of values observed (Fig 4) [29]. Nevertheless, this finding is in agreement with the similarly weak correspondence between feather δ$^{18}$O and δD values and local meteoric water lines observed in previous studies [1, 10, 32]. Broadly, this observation implies that δ$^{18}$O and δD values of these feathers may be more strongly related to diet (as with δ$^{13}$C and δ$^{15}$N values) than with local hydrology. Isotopic enrichment factors for δ$^{18}$O and δD isotopes in the *S. citrina* diet would presumably play a significant role in the isotopic signatures of their feathers. We note that Wolf et al. (2012) [34] found the relationship between diet and δD and δ$^{18}$O values of feathers was weak in feathers for the Japanese quail, indicating that further study on these isotope systems is needed.

Finally, we note that despite the general isotopic homogeneity, there were a few outliers. In particular, δ$^{13}$C and δ$^{15}$N values of rectrices RL2 for the 2018 bird are highly anomalous despite their δ$^{18}$O or δD values being within the range of the other feathers (Fig 3). Normal δ$^{18}$O and δD values seem to suggest that these feathers were still molted on or near breeding grounds, but anomalous δ$^{13}$C and δ$^{15}$N values imply that these feathers grew under different

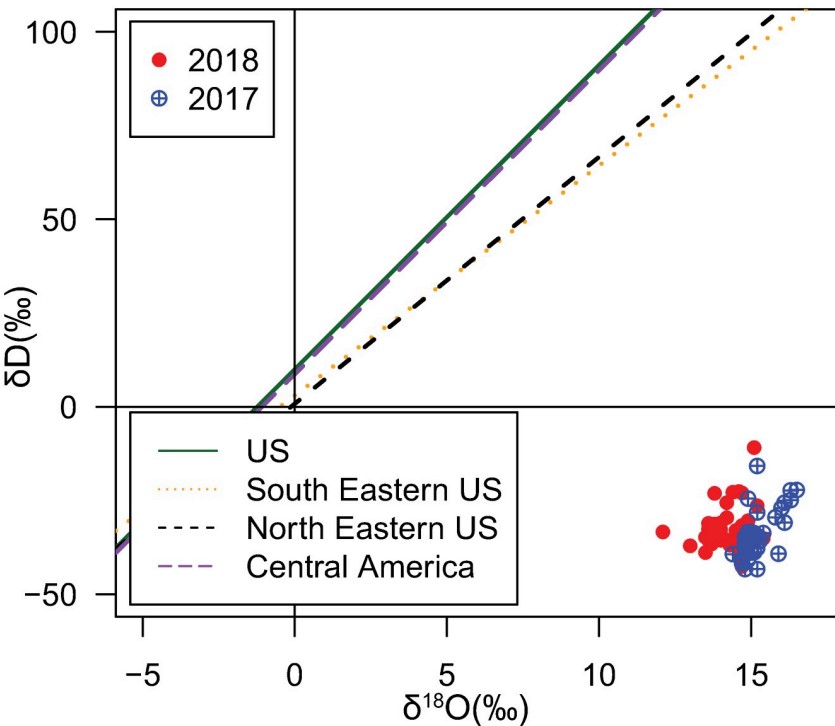

**Fig 4. Comparison of the δ$^{18}$O vs. δD for the 2017 and 2018 feather sets and local meteoric waterlines.** The meteoric water lines included are the contiguous United States (solid, green), Southeastern United States (dotted, orange), Northeastern United States (short dash, black), and Central America (long dash, purple) [29, 35].

circumstances than the others. One possibility is that these outlier feathers represent replacement feathers after feather damage or loss, although we did not observe any molt limits that might clearly support such a hypothesis.

## Conclusion

Oxygen, hydrogen, carbon and nitrogen isotopic signatures of the feathers from two Hooded Warblers (*S. citrina*) show striking within-bird homogeneity, confirming the hypothesis that molt in these neotropical migrants is fully completed on the breeding grounds. The homogeneity within single birds also suggests that single-feather isotopic studies should be generally robust for isotopic fingerprinting within this species. However, further study should be performed on other birds that undergo molt during migration to confirm whether single feather analyses can be used. Finally, despite this homogeneity, there are occasional outlier feathers, which could signify regrowth of lost feathers. When undergoing feather analysis, the possibility of these anomalous feathers should be considered.

## Supporting information

**S1 File. Values for $\delta^{13}C$, $\delta^{15}N$, $\delta^{18}O$, and $\delta D$ for each bird sample by feather type.** (XLSX)

## Author Contributions

**Conceptualization:** Samiksha Deme, Laurence Y. Yeung, Tao Sun, Cin-Ty A. Lee.

**Data curation:** Samiksha Deme, Tao Sun.

**Formal analysis:** Samiksha Deme.

**Investigation:** Samiksha Deme, Laurence Y. Yeung, Tao Sun, Cin-Ty A. Lee.

**Methodology:** Samiksha Deme, Laurence Y. Yeung, Tao Sun, Cin-Ty A. Lee.

**Resources:** Laurence Y. Yeung, Tao Sun, Cin-Ty A. Lee.

**Supervision:** Laurence Y. Yeung, Tao Sun, Cin-Ty A. Lee.

**Writing – original draft:** Samiksha Deme, Laurence Y. Yeung, Cin-Ty A. Lee.

**Writing – review & editing:** Samiksha Deme, Laurence Y. Yeung, Cin-Ty A. Lee.

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
