## [Decision Letter · Decision Letter 0]

30 Sep 2020

PONE-D-20-20872

Stable isotope (C,N,O, and H) study of a comprehensive set of feathers from two Setophaga citrina

PLOS ONE

Dear Dr. Deme,

Thank you for submitting your manuscript to PLOS ONE. After careful consideration, we feel that it has merit but does not fully meet PLOS ONE’s publication criteria as it currently stands. Therefore, we invite you to submit a revised version of the manuscript that addresses the points raised during the review process.

We look forward to receiving your revised manuscript.

Kind regards,

Huan Cui, Ph.D.

Academic Editor

PLOS ONE

2. We note that Figure 1 in your submission contain map images which may be copyrighted. All PLOS content is published under the Creative Commons Attribution License (CC BY 4.0), which means that the manuscript, images, and Supporting Information files will be freely available online, and any third party is permitted to access, download, copy, distribute, and use these materials in any way, even commercially, with proper attribution. For these reasons, we cannot publish previously copyrighted maps or satellite images created using proprietary data, such as Google software (Google Maps, Street View, and Earth). For more information, see our copyright guidelines: http://journals.plos.org/plosone/s/licenses-and-copyright.

2.1.    You may seek permission from the original copyright holder of Figure 1 to publish the content specifically under the CC BY 4.0 license. 

2.2.    If you are unable to obtain permission from the original copyright holder to publish these figures under the CC BY 4.0 license or if the copyright holder’s requirements are incompatible with the CC BY 4.0 license, please either i) remove the figure or ii) supply a replacement figure that complies with the CC BY 4.0 license. Please check copyright information on all replacement figures and update the figure caption with source information. If applicable, please specify in the figure caption text when a figure is similar but not identical to the original image and is therefore for illustrative purposes only.

Reviewers' comments:

Reviewer's Responses to Questions

**Comments to the Author**

1. Is the manuscript technically sound, and do the data support the conclusions?

Reviewer #1: Yes

Reviewer #2: Yes

2. Has the statistical analysis been performed appropriately and rigorously? 

Reviewer #1: Yes

Reviewer #2: Yes

3. Have the authors made all data underlying the findings in their manuscript fully available?

Reviewer #1: Yes

Reviewer #2: Yes

4. Is the manuscript presented in an intelligible fashion and written in standard English?

Reviewer #1: Yes

Reviewer #2: Yes

5. Review Comments to the Author

Reviewer #1: Deme and co-authors look at the isotopic variability among feathers and body parts of Hooded Warblers to determine if there is consistent variability associated with the pattern of molting, suggesting that molting occurs not only on the breeding grounds. They find that—for the most part—all feathers record the same isotopic values, indicating that the molt is fully completed on the breeding grounds. As a consequence, single feathers can more confidently be used to isotopically finger-print the breeding locations.

This is a nice clean study, with a dataset that nicely answers the question put forward. The figures support the conclusions and the citations are appropriate. Overall, I think this manuscript can be published with only minor revisions. I should note that I am not an expert on bird ecology or migration patterns (though Hooded Warblers are some of my favorite warblers) nor am I an expert on isotopes in ecology. Nevertheless, I use light stable isotopes regularly in my research, particularly d13C, dD, and d18O as a way of understanding the natural environment.

I have three main suggestions for revisions. I think all of these will simply require some additional writing in the text, rather than any new laboratory analyses.

1. First, there needs to be more description about how the birds were stored prior to analysis. There appears to be some amount of relatively fast (ie, less than 2 weeks) H-isotope exchange between the keratin and water (Chamberlain et al., 1997). Chamberlain et al. (1997) conclude that only about 13% of the hydrogen is truly exchangeable; thus, the effect is likely to be small, but depends on the ambient water vapor dD and the dD values of the feathers when the warblers were collected. Given that there isn’t tremendous variability in dD between winter-time Central America, summertime along the Eastern Seaboard, and Houston air, I would further expect this effect to be small. Nevertheless, a simple calculation to show what the possible maximum effect of exchange would likely be (due to storage or delayed analysis) would help to clarify the meaning of the dD data.

2. The authors seem to somewhat “punt” and not actually dive into the d18O and dD data as much as one would like. First, it would be helpful to plot the feather dD against the feather d18O in order to clearly demonstrate that this relationship does not follow the global or really any meteoric water line. Second, the lack of a correspondence with an MWL doesn’t seem too surprising as Hobson et al. (2004) show that there is a poor correlation between feather d18O and local water d18O. Thus, it may be that only the dD is a tracer of local water. Here, though, I think the authors can again do a more thorough analysis than currently presented. Both Chamberlain et al. (1997) and Hobson et al. (2004) show that feather dD is depleted relative to meteoric water by about 10‰. In this study, that would suggest that the breeding ground meteoric water is in the range of -20 to -30‰. This does appear to be substantially higher than the dD values of meteoric waters in the Northeast and even down to the Potomac; it is, however, more in line with meteoric water dD values in the southeast (Kendall and Coplen, 2001). I don’t know what wintertime dD values are in Central America, but this can probably be approximated by searching waterisotopes.org (https://wateriso.utah.edu/waterisotopes/pages/spatial_db/SPATIAL_DB.html). I’d guess that something more specific can be said about the source of the water based upon the dD values even if no meteoric water line can be recovered from the feather d18O and dD.

3. The authors find that the d13C values are approximately -25‰ and suggest that there is an approximate enrichment of 3.5‰ from what the birds are eating, suggesting that the vegetation in the breeding grounds has a d13C of -28.5‰. However, my understanding is that these birds are insectivores, which likely additionally fractionate 13C (say 1 or 1.5‰), indicating that the base vegetation must be even lower in d13C (~ -29.5 or -30‰). This isn’t crazy low…instead, it might indicate that the breeding ground vegetation receives fairly high mean annual precipitation (> 1000 mm/year) (Kohn, 2010), perhaps further indicative that the breeding grounds are in the Southeast, or that the insects consumed by Hooded Warblers are targeting specific leaves that have lower d13C. Regardless, I’m not sure why the insect dietary fractionation of 13C can be excluded when calculating the original vegetation component.

In conclusion, I think the above comments will not be difficult to address, but will help to add some richness to the very nice dataset that is presented here.

Jeremy Rugenstein

Colorado State University

References cited in review:

Chamberlain, C.P., Blum, J.D., Holmes, R.T., Feng, X., Sherry, T.W., Graves, G.R., 1997. The use of isotope tracers for identifying populations of migratory birds. Oecologia 109, 132–141.

Hobson, K.A., Bowen, G.J., Wassenaar, L.I., Ferrand, Y., Lormee, H., 2004. Using stable hydrogen and oxygen isotope measurements of feathers to infer geographical origins of migrating European birds. Oecologia 141, 477–488. https://doi.org/10.1007/s00442-004-1671-7

Kendall, C., Coplen, T.B., 2001. Distribution of oxygen-18 and deuterium in river waters across the United States. Hydrol. Process. 15, 1363–1393. https://doi.org/10.1002/hyp.217

Kohn, M.J., 2010. Carbon isotope compositions of terrestrial C3 plants as indicators of (paleo)ecology and (paleo)climate. Proc. Natl. Acad. Sci. U. S. A. 107, 19691–19695. https://doi.org/10.1073/pnas.1004933107

Reviewer #2: Deme et al. investigate molt strategy and timing using two specimens of S. citrina from two different spring seasons, and several stable isotope systems. This is used to assess the robustness of using single feathers for reconstructing ecological behaviour and migratory patterns of birds. While it is a small data set, some differences and outliers need to be explored in more detail in further studies, and no reference material of local water, insects were included, the hypothesis is compelling and and the discussion is supported by the available results. However, I recommend revisions (see comments below) before publication. Also, I recommend emphasising that further analysis is needed, for example in your abstract; your title could also emphasis that this is a first tentative evaluation of a full set of feathers.

Comments:

Explain the used stable isotope systems and why they make sense for your study in more detail.

Consider including other studies of comprehensive sets of feathers and their results in context of your study. For example, the work by English et al. 2018 on museum specimens.

Line 57: Is it known why these birds died? Where they both male or female, or one of each? Would a sickness potentially cause changes to feeding behaviour and/or effect any of the used isotope systems?

Line 68: Analog to line 59, include the feather IDs used in Fig. 2 in brackets for those, who cannot print in colour.

Line 83: Too long, shorten. Also, how often were the international standards analysed? Include n.

Line 87: Why would you analyse your samples in duplicate or triplicate to evaluate external reproducibility? Should this not be evaluated using reference materials?

Line 97: Which one is it, n=2 or 3? If RL2 was replicated twice and S3 three times, then it would be better to write this as: … of the anomalous feathers RL2 and S3 (n= 2 and 3, respectively) suggest…

Line 106: Here and elsewhere, when giving mean values and the p-value, please include n.

Line 109: The error range of your mean values when comparing the 2017 vs 2018 bird, are app. equal (d15N) or even higher (dD) than the mean value – this error range seems to undermine the usefulness of a mean value. Perhaps reconsider what this range and the inferred difference between bird 2017 and 2018 means.

Line 119: It is interesting that r2 for d13C are more often in disagreement between bird 2017 and 2018; particularly S1 to P9, S1 to S7, and RL1 to RL6 show a high r2 for bird 2017, while the r2 values show no correlation for bird 2018. When you highlight one high correlation for d18O, RR1 to RR6 for bird 2017, consider evaluating Table 1 in more detail.

Line 140: Expand on your reasoning for this.

Line 165: What do you mean with ‘rare’ oxygen and hydrogen isotopes? It would be better to refer to, e.g., C-13 as less abundant.

Reference:

English Ph. A., Green D. J., and Nocera J.J., 2018. Stable Isotopes from Museum Specimens May Provide Evidence of Long-Term Change in the Trophic Ecology of a Migratory Aerial Insectivore. Front. Ecol. Evol., https://doi.org/10.3389/fevo.2018.00014.

6. PLOS authors have the option to publish the peer review history of their article (what does this mean?). If published, this will include your full peer review and any attached files.

Reviewer #1: **Yes: **Jeremy Rugenstein

Reviewer #2: No

---

## [Author Response · Author response to Decision Letter 0]

1 Dec 2020

We thank the reviewers for their comments on the paper and the time they took to edit this work. Specific responses are in the file "Response to Reviewers" document uploaded previously.

---

## [Decision Letter · Decision Letter 1]

26 Dec 2020

Stable isotope (C,N,O, and H) study of a comprehensive set of feathers from two Setophaga citrina

PONE-D-20-20872R1

Dear Dr. Deme,

We’re pleased to inform you that your manuscript has been judged scientifically suitable for publication and will be formally accepted for publication once it meets all outstanding technical requirements.

Kind regards,

Huan Cui, Ph.D.

Academic Editor

PLOS ONE

Additional Editor Comments (optional):

Reviewers' comments:

Reviewer's Responses to Questions

**Comments to the Author**

1. If the authors have adequately addressed your comments raised in a previous round of review and you feel that this manuscript is now acceptable for publication, you may indicate that here to bypass the “Comments to the Author” section, enter your conflict of interest statement in the “Confidential to Editor” section, and submit your "Accept" recommendation.

Reviewer #1: All comments have been addressed

Reviewer #2: All comments have been addressed

2. Is the manuscript technically sound, and do the data support the conclusions?

Reviewer #1: Yes

Reviewer #2: Yes

3. Has the statistical analysis been performed appropriately and rigorously? 

Reviewer #1: Yes

Reviewer #2: Yes

4. Have the authors made all data underlying the findings in their manuscript fully available?

Reviewer #1: Yes

Reviewer #2: Yes

5. Is the manuscript presented in an intelligible fashion and written in standard English?

Reviewer #1: Yes

Reviewer #2: Yes

6. Review Comments to the Author

Reviewer #1: Thanks for addressing my earlier comments. I found the meteoric water line comparison quite interesting, and it does seem like the variability in d18O is somewhat compressed relative to what one might expect if O and H isotopes were more directly reflecting meteoric waters.

Reviewer #2: I propose accepting this manuscript for publication, however, consider a few minor corrections:

L11: "...of a single feather may be representative..."

L54: "...by the work of Graves et al. (2018), ... and timing in two after hatch year (AHY) male ..."

L95: parallel to e.g. L84, use round brackets: "Primary (P) and secondary (S) flight feathers..."

7. PLOS authors have the option to publish the peer review history of their article (what does this mean?). If published, this will include your full peer review and any attached files.

Reviewer #1: **Yes: **Jeremy K C Rugenstein

Reviewer #2: No

---

## [Editor Report · Acceptance letter]

30 Dec 2020

PONE-D-20-20872R1 

Stable isotope (C,N,O, and H) study of a comprehensive set of feathers from two *Setophaga citrina*

Dear Dr. Deme:

I'm pleased to inform you that your manuscript has been deemed suitable for publication in PLOS ONE. Congratulations! Your manuscript is now with our production department. 

Kind regards, 

on behalf of

Dr. Huan Cui 

Academic Editor

PLOS ONE